# ANIMATEDIFF: ANIMATE YOUR PERSONALIZED TEXT-TO-IMAGE DIFFUSION MODELS WITHOUT SPECIFIC TUNING

**Yuwei Guo**[1]   **Ceyuan Yang**[2][†]  **Anyi Rao**[3]   **Zhengyang Liang**[2]   **Yaohui Wang**[2]
**Yu Qiao**[2]   **Maneesh Agrawala**[3]   **Dahua Lin**[1,2]   **Bo Dai**[2]
[1]The Chinese University of Hong Kong   [2]Shanghai Artificial Intelligence Laboratory
[3]Stanford University

*(cartoon) 1boy, dark skin, playing guitar, concert, . . .*   *(oil painting) black pearl pirate ship, night time, sea, . . .*   *(realistic) a Lamborghini on road, fireworks, high detail, . . .*

*zoom-in*       *rolling*       *zoom-out + rolling*       *right + up*

Figure 1: AnimateDiff directly turns existing personalized text-to-image (T2I) models to the corresponding animation generators with a pre-trained motion module. *First row*: results by combining AnimateDiff with three personalized T2Is in different domains; *Second row*: results of further combining AnimateDiff with MotionLoRA (s) to achieve shot type controls. *Best viewed with Acrobat Reader. Click the images to play the animation clips.*

## ABSTRACT

With the advance of text-to-image (T2I) diffusion models (*e.g.*, Stable Diffusion) and corresponding personalization techniques such as DreamBooth and LoRA, everyone can manifest their imagination into high-quality images at an affordable cost. However, adding motion dynamics to existing high-quality personalized T2Is and enabling them to generate animations remains an open challenge. In this paper, we present AnimateDiff, a practical framework for animating personalized T2I models without requiring model-specific tuning. At the core of our framework is a plug-and-play motion module that can be trained once and seamlessly integrated into any personalized T2Is originating from the same base T2I. Through our proposed training strategy, the motion module effectively learns transferable motion priors from real-world videos. Once trained, the motion module can be inserted into a personalized T2I model to form a personalized animation

---

[†]Corresponding Author.

generator. We further propose MotionLoRA, a lightweight fine-tuning technique for AnimateDiff that enables a pre-trained motion module to adapt to new motion patterns, such as different shot types, at a low training and data collection cost. We evaluate AnimateDiff and MotionLoRA on several public representative personalized T2I models collected from the community. The results demonstrate that our approaches help these models generate temporally smooth animation clips while preserving the visual quality and motion diversity. Codes and pre-trained weights are available at `https://github.com/guoyww/AnimateDiff`.

# 1 INTRODUCTION

Text-to-image (T2I) diffusion models (Nichol et al., 2021; Ramesh et al., 2022; Saharia et al., 2022; Rombach et al., 2022) have greatly empowered artists and amateurs to create visual content using text prompts. To further stimulate the creativity of existing T2I models, lightweight personalization methods, such as DreamBooth (Ruiz et al., 2023) and LoRA (Hu et al., 2021) have been proposed. These methods enable customized fine-tuning on small datasets using consumer-grade hardware such as a laptop with an RTX3080, thereby allowing users to adapt a base T2I model to new domains and improve visual quality at a relatively low cost. Consequently, a large community of AI artists and amateurs has contributed numerous personalized models on model-sharing platforms such as Civitai (2022) and Hugging Face (2022). While these personalized T2I models can generate remarkable visual quality, their outputs are limited to static images. On the other hand, the ability to generate animations is more desirable in real-world production, such as in the movie and cartoon industries. In this work, we aim to directly transform existing high-quality personalized T2I models into animation generators without requiring model-specific fine-tuning, which is often impractical in terms of computation and data collection costs for amateur users.

We present AnimateDiff, an effective pipeline for addressing the problem of animating personalized T2Is while preserving their visual quality and domain knowledge. The core of AnimateDiff is an approach for training a plug-and-play motion module that learns reasonable motion priors from video datasets, such as WebVid-10M (Bain et al., 2021). At inference time, the trained motion module can be directly integrated into personalized T2Is and produce smooth and visually appealing animations without requiring specific tuning. The training of the motion module in AnimateDiff consists of three stages. Firstly, we fine-tune a domain adapter on the base T2I to align with the visual distribution of the target video dataset. This preliminary step guarantees the motion module concentrates on learning the motion priors rather than pixel-level details from the training videos. Secondly, we inflate the base T2I together with the domain adapter and introduce a newly initialized motion module for motion modeling. We then optimize this module on videos while keeping the domain adapter and base T2I weights fixed. By doing so, the motion module learns generalized motion priors and can, via module insertion, enable other personalized T2Is to generate smooth and appealing animations aligned with their personalized domains. The third stage of AnimateDiff, also dubbed as MotionLoRA, aims to adapt the pre-trained motion module to specific motion patterns with a small number of reference videos and training iterations. We achieve this by fine-tuning the motion module with the aid of Low-Rank Adaptation (LoRA) (Hu et al., 2021). Remarkably, adapting to a new motion pattern can be achieved with as few as 50 reference videos. Moreover, a MotionLoRA model requires only approximately 30M of additional storage space, further enhancing the efficiency of model sharing. This efficiency is particularly valuable for users who are unable to bear the expensive costs of pre-training but desire to fine-tune the motion module for specific effects.

We evaluate the performance of AnimateDiff and MotionLoRA on a diverse set of personalized T2I models collected from model-sharing platforms (Civitai, 2022; Hugging Face, 2022). These models encompass a wide spectrum of domains, ranging from 2D cartoons to realistic photographs, thereby forming a comprehensive benchmark for our evaluation. The results of our experiments demonstrate promising outcomes. In practice, we also found that a Transformer (Vaswani et al., 2017) architecture along the temporal axis is adequate for capturing appropriate motion priors. We also demonstrate that our motion module can be seamlessly integrated with existing content-controlling approaches (Zhang et al., 2023; Mou et al., 2023) such as ControlNet without requiring additional training, enabling AnimateDiff for controllable animation generation.

In summary, (1) we present AnimateDiff, a practical pipeline that enables the animation generation ability of any personalized T2Is without specific fine-tuning; (2) we verify that a Transformer architecture is adequate for modeling motion priors, which provides valuable insights for video generation; (3) we propose MotionLoRA, a lightweight fine-tuning technique to adapt pre-trained motion modules to new motion patterns; (4) we comprehensively evaluate our approach with representative community models and compare it with both academic baselines and commercial tools such as Gen-2 (2023) and Pika Labs (2023). Furthermore, we showcase its compatibility with existing works for controllable generation.

## 2    RELATED WORK

**Text-to-image diffusion models.** Diffusion models (Ho et al., 2020; Dhariwal & Nichol, 2021; Song et al., 2020) for text-to-image (T2I) generation (Gu et al., 2022; Mokady et al., 2023; Podell et al., 2023; Ding et al., 2021; Zhou et al., 2022b; Ramesh et al., 2021; Li et al., 2022) have gained significant attention in both academic and non-academic communities recently. GLIDE (Nichol et al., 2021) introduced text conditions and demonstrated that incorporating classifier guidance leads to more pleasing results. DALL-E2 (Ramesh et al., 2022) improves text-image alignment by leveraging the CLIP (Radford et al., 2021) joint feature space. Imagen (Saharia et al., 2022) incorporates a large language model (Raffel et al., 2020) and a cascade architecture to achieve photorealistic results. Latent Diffusion Model (Rombach et al., 2022), also known as Stable Diffusion, moves the diffusion process to the latent space of an auto-encoder to enhance efficiency. eDiff-I (Balaji et al., 2022) employs an ensemble of diffusion models specialized for different generation stages.

**Personalizing T2I models.** To facilitate the creation with pre-trained T2Is, many works focus on efficient model personalization (Shi et al., 2023; Lu et al., 2023; Dong et al., 2022; Kumari et al., 2023), *i.e.*, introducing concepts or styles to the base T2I using reference images. The most straightforward approach to achieve this is complete fine-tuning of the model. Despite its potential to significantly enhance overall quality, this practice can lead to catastrophic forgetting (Kirkpatrick et al., 2017; French, 1999) when the reference image set is small. Instead, DreamBooth (Ruiz et al., 2023) fine-tunes the entire network with preservation loss and uses only a few images. Textual Inversion (Gal et al., 2022) optimize a token embedding for each new concept. Low-Rank Adaptation (LoRA) (Hu et al., 2021) facilitates the above fine-tuning process by introducing additional LoRA layers to the base T2I and optimizing only the weight residuals. There are also encoder-based approaches that address the personalization problem (Gal et al., 2023; Jia et al., 2023). In our work, we focus on tuning-based methods, including overall fine-tuning, DreamBooth (Ruiz et al., 2023), and LoRA (Hu et al., 2021), as they preserve the original feature space of the base T2I.

**Animating personalized T2Is.** There are not many existing works regarding animating personalized T2Is. Text2Cinemagraph (Mahapatra et al., 2023) proposed to generate cinematography via flow prediction. In the field of video generation, it is common to extend a pre-trained T2I with temporal structures. Existing works (Esser et al., 2023; Zhou et al., 2022a; Singer et al., 2022; Ho et al., 2022b,a; Ruan et al., 2023; Luo et al., 2023; Yin et al., 2023b,a; Wang et al., 2023b; Hong et al., 2022; Luo et al., 2023) mostly update all parameters and modify the feature space of the original T2I and is not compatible with personalized ones. Align-Your-Latents (Blattmann et al., 2023) shows that the frozen image layers in a general video generator can be personalized. Recently, some video generation approaches have shown promising results in animating a personalized T2I model. Tune-a-Video (Wu et al., 2023) fine-tune a small number of parameters on a single video. Text2Video-Zero (Khachatryan et al., 2023) introduces a training-free method to animate a pre-trained T2I via latent wrapping based on a pre-defined affine matrix.

## 3    PRELIMINARY

We introduce the preliminary of Stable Diffusion (Rombach et al., 2022), the base T2I model used in our work, and Low-Rank Adaptation (LoRA) (Hu et al., 2021), which helps understand the domain adapter (Sec. 4.1) and MotionLoRA (Sec. 4.3) in AnimateDiff.

**Stable Diffusion.** We chose Stable Diffusion (SD) as the base T2I model in this paper since it is open-sourced and has a well-developed community with many high-quality personalized T2I models for evaluation. SD performs the diffusion process within the latent space of a pre-trained autoen-

coder $\mathcal{E}(\cdot)$ and $\mathcal{D}(\cdot)$. In training, an encoded image $z_0 = \mathcal{E}(x_0)$ is perturbed to $z_t$ by the forword diffusion:

$$z_t = \sqrt{\bar{\alpha}_t} z_0 + \sqrt{1 - \bar{\alpha}_t} \epsilon, \ \epsilon \sim \mathcal{N}(0, I), \tag{1}$$

for $t = 1, \ldots, T$, where pre-defined $\bar{\alpha}_t$ determines the noise strength at step $t$. The denoising network $\epsilon_\theta(\cdot)$ learns to reverse this process by predicting the added noise, encouraged by an MSE loss:

$$\mathcal{L} = \mathbb{E}_{\mathcal{E}(x_0), y, \epsilon \sim \mathcal{N}(0, I), t} \left[ \| \epsilon - \epsilon_\theta(z_t, t, \tau_\theta(y)) \|_2^2 \right], \tag{2}$$

where $y$ is the text prompt corresponding to $x_0$; $\tau_\theta(\cdot)$ is a text encoder mapping the prompt to a vector sequence. In SD, $\epsilon_\theta(\cdot)$ is implemented as a UNet (Ronneberger et al., 2015) consisting of pairs of down/up sample blocks at four resolution levels, as well as a middle block. Each network block consists of ResNet (He et al., 2016), spatial self-attention layers, and cross-attention layers that introduce text conditions.

**Low-rank adaptation (LoRA).** LoRA (Hu et al., 2021) is an approach that accelerates the fine-tuning of large models and is first proposed for language model adaptation. Instead of retraining all model parameters, LoRA adds pairs of rank-decomposition matrices and optimizes only these newly introduced weights. By limiting the trainable parameters and keeping the original weights frozen, LoRA is less likely to cause catastrophic forgetting (Kirkpatrick et al., 2017). Concretely, the rank-decomposition matrices serve as the residual of the pre-trained model weights $\mathcal{W} \in \mathbb{R}^{m \times n}$. The new model weight with LoRA is

$$\mathcal{W}' = \mathcal{W} + \Delta \mathcal{W} = \mathcal{W} + AB^T, \tag{3}$$

where $A \in \mathbb{R}^{m \times r}$, $B \in \mathbb{R}^{n \times r}$ are a pair of rank-decomposition matrices, $r$ is a hyper-parameter, which is referred to as the rank of LoRA layers. In practice, LoRA is only applied to attention layers, further reducing the cost and storage for model fine-tuning.

# 4 ANIMATEDIFF

The core of our method is learning transferable motion priors from video data, which can be applied to personalized T2Is without specific tuning. As shown in Fig. 2, at inference time, our motion module (blue) and the *optional* MotionLoRA (green) can be directly inserted into a personalized T2I to constitute the animation generator, which subsequently generates animations via an iterative denoising process.

We achieve this by training three components of AnimateDiff, namely domain adapter, motion module, and MotionLoRA. The domain adapter in Sec. 4.1 is only used in the training to alleviate the negative effects caused by the visual distribution gap between the base T2I pre-training data and our video training data; the motion module in Sec. 4.2 is for learning the motion

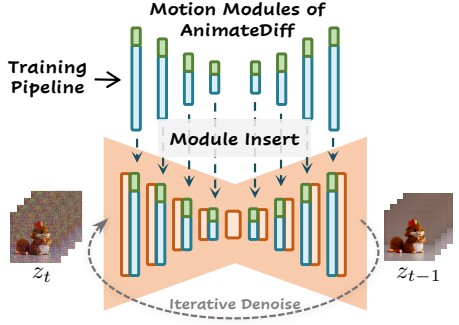

Figure 2: Inference pipeline.

priors; and the MotionLoRA in Sec. 4.3, which is *optional* in the case of general animation, is for adapting pre-trained motion modules to new motion patterns. Sec.4.4 elaborates on the training (Fig. 3) and inference of AnimateDiff.

## 4.1 ALLEVIATE NEGATIVE EFFECTS FROM TRAINING DATA WITH DOMAIN ADAPTER

Due to the difficulty in collection, the visual quality of publicly available video training datasets is much lower than their image counterparts. For example, the contents of the video dataset We-bVid (Bain et al., 2021) are mostly real-world recordings, whereas the image dataset LAION-Aesthetic (Schuhmann et al., 2022) contains higher-quality contents, including artistic paintings and professional photography. Moreover, when treated individually as images, each video frame can contain motion blur, compression artifacts, and watermarks. Therefore, there is a non-negligible quality domain gap between the high-quality image dataset used to train the base T2I and the target

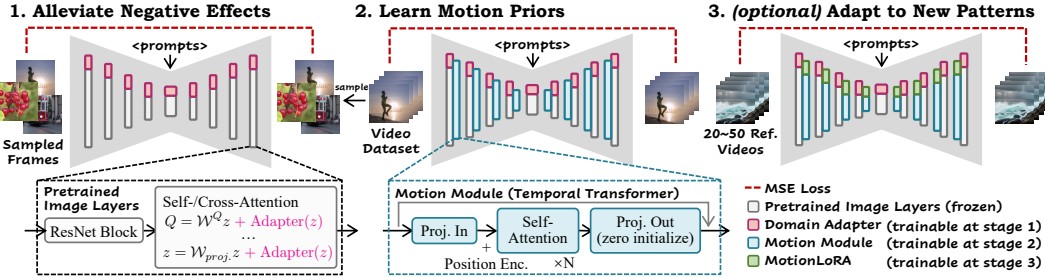

Figure 3: **Training pipeline of AnimateDiff.** AnimateDiff consists of three training stages for the corresponding component modules. Firstly, a domain adapter (Sec. 4.1) is trained to alleviate the negative effects caused by training videos. Secondly, a motion module (Sec. 4.2) is inserted and trained on videos to learn general motion priors. Lastly, MotionLoRA (Sec. 4.3) is trained on a few reference videos to adapt the pre-trained motion module to new motion patterns.

video dataset we use for learning the motion priors. We argue that such a gap can limit the quality of the animation generation pipeline when trained directly on the raw video data.

To avoid learning this quality discrepancy as part of our motion module and preserve the knowledge of the base T2I, we propose to **fit the domain information to a separate network**, dubbed as domain adapter. We drop the domain adapter at inference time and show that this practice helps reduce the negative effects caused by the domain gap mentioned above. We implement the domain adapter layers with LoRA (Hu et al., 2021) and insert them into the self-/cross-attention layers in the base T2I, as shown in Fig. 3. Take query (Q) projection as an example. The internal feature $z$ after projection becomes

$$Q = \mathcal{W}^Q z + \text{AdapterLayer}(z) = \mathcal{W}^Q z + \alpha \cdot AB^T z, \tag{4}$$

where $\alpha = 1$ is a scalar and can be adjusted to other values at inference time (set to $0$ to remove the effects of domain adapter totally). We then optimize only the parameters of the domain adapter on static frames randomly sampled from video datasets with the same objective in Eq. (2).

## 4.2 LEARN MOTION PRIORS WITH MOTION MODULE

To model motion dynamics along the temporal dimension on top of a pre-trained T2I, we must 1) inflate the 2-dimensional diffusion model to deal with 3-dimensional video data and 2) design a sub-module to enable efficient information exchange along the temporal axis.

**Network Inflation.** The pre-trained image layers in the base T2I model capture high-quality content priors. To utilize the knowledge, a preferable way for network inflation is to let these image layers independently deal with video frames. To achieve this, we adopt a practice similar to recent works (Ho et al., 2022b; Wu et al., 2023; Blattmann et al., 2023), and modify the model so that it takes 5D video tensors $x \in \mathbb{R}^{b \times c \times f \times h \times w}$ as input, where $b$ and $f$ represent batch axis and frame-time axis respectively. When the internal feature maps go through image layers, the temporal axis $f$ is ignored by being reshaped into the $b$ axis, allowing the network to process each frame independently. We then reshape the feature map to the 5D tensor after the image layer. On the other hand, our newly inserted motion module ignores the spatial axis by reshaping $h, w$ into $b$ and then reshaping back after the module.

**Module Design.** Recent works on video generation have explored many designs for temporal modeling. In AnimateDiff, we adopt the Transformer (Vaswani et al., 2017) architecture as our motion module design, and make minor modifications to adapt it to operate along the temporal axis, which we refer to as "temporal Transformer" in the following sections. We experimentally found this design is adequate for modeling motion priors. As illustrated in Fig. 3, the temporal Transformer consists of several self-attention blocks along the temporal axis, with sinusoidal position encoding to encode the location of each frame in the animation. As mentioned above, the input of the motion module is the reshaped feature map whose spatial dimensions are merged into the batch axis. When we divide the reshaped feature map along the temporal axis, it can be regarded as vector sequences with length of $f$, *i.e.*, $\{z_1, ..., z_f; z_i \in \mathbb{R}^{(b \times h \times w) \times c}\}$. The vectors will then be projected and go

through several self-attention blocks, *i.e.*

$$z_{out} = \text{Attention}(Q, K, V) = \text{Softmax}(QK^T/\sqrt{c}) \cdot V, \tag{5}$$

where $Q = W^Q z$, $K = W^K z$, and $V = W^V z$ are three separated projections. The attention mechanism enables the generation of the current frame to incorporate information from other frames. As a result, instead of generating each frame individually, the T2I model inflated with our motion module learns to capture the changes of visual content over time, which constitute the motion dynamics in an animation clip. Note that sinusoidal position encoding added before the self-attention is essential; otherwise, the module is not aware of the frame order in the animation. To avoid any harmful effects that the additional module might introduce, we zero initialize (Zhang et al., 2023) the output projection layers of the temporal Transformer and add a residual connection so that the motion module is an identity mapping at the beginning of training.

### 4.3 ADAPT TO NEW MOTION PATTERNS WITH MOTIONLORA

While the pre-trained motion module captures general motion priors, a question arises when we need to effectively adapt it to new motion patterns such as camera zooming, panning and rolling, *etc.*, with a small number of reference videos and training iterations. Such efficiency is essential for users who cannot afford expensive pre-training costs but would like to fine-tune the motion module for specific effects. Here comes the last stage of AnimateDiff, also dubbed as MotionLoRA (Fig. 3), an efficient fine-tuning approach for **motion personalization**. Considering the architecture of the motion module and the limited number of reference videos, we add LoRA layers to the self-attention layers of the motion module in the inflated model described in Sec. 4.2, then train these LoRA layers on the reference videos of new motion patterns.

We experiment with several shot types and get the reference videos via rule-based data augmentation. For instance, to get videos with zooming effects, we augment the videos by gradually reducing (zoom-in) or enlarging (zoom-out) the cropping area of video frames along the temporal axis. We demonstrate that our MotionLoRA can achieve promising results even with as few as $20 \sim 50$ reference videos, 2,000 training iterations (around $1 \sim 2$ hours) as well as about 30M storage space, enabling efficient model tuning and sharing among users. Benefited by the low-rank property, MotionLoRA also has the composition capability. Namely, individually trained MotionLoRA models can be combined to achieve composed motion effects at inference time.

### 4.4 ANIMATEDIFF IN PRACTICE

**Training.** As illustrated in Fig. 3, AnimateDiff consists of three trainable component modules to learn transferable motion priors. Their training objectives are slightly different. The domain adapter is trained with the original objective as in Eq. (2). The motion module and MotionLoRA, as part of an animation generator, use a similar objective with minor modifications to accommodate higher dimension video data. Concretely, a video data batch $x_0^{1:f} \in \mathbb{R}^{b \times c \times f \times h \times w}$ is first encoded into the latent codes $z_0^{1:f}$ frame-wisely via the pre-trained auto-encoder of SD. The latent codes are then noised using the defined forward diffusion schedule as in Eq. (1)

$$z_t^{1:f} = \sqrt{\bar{\alpha}_t} z_0^{1:f} + \sqrt{1 - \bar{\alpha}_t} \epsilon^{1:f}. \tag{6}$$

The inflated model inputs the noised latent codes and corresponding text prompts and predicts the added noises. The final training objective of our motion modeling module is:

$$\mathcal{L} = \mathbb{E}_{\mathcal{E}(x_0^{1:f}), y, \epsilon^{1:f} \sim \mathcal{N}(0, I), t} \left[ \| \epsilon - \epsilon_\theta(z_t^{1:f}, t, \tau_\theta(y)) \|_2^2 \right]. \tag{7}$$

It's worth noting that when training the domain adapter, the motion module, and the MotionLoRA, parameters outside the trainable part remain frozen.

**Inference.** At inference time (Fig. 2), the personalized T2I model will first be inflated in the same way discussed in Section 4.2, then injected with the motion module for general animation generation, and the *optional* MotionLoRA for generating animation with personalized motion. As for the domain adapter, instead of simply dropping it during the inference time, in practice, we can also inject it into the personalized T2I model and adjust its contribution by changing the scaler $\alpha$ in Eq. (4). An ablation study on the value of $\alpha$ is conducted in experiments. Finally, the animation frames can be obtained by performing the reverse diffusion process and decoding the latent codes.

*RCNZ Cartoon 3d*      *TUSUN*      *epiC Realism*      *ToonYou*

*a golden Labrador, natural lighting, . . .*    *cute Pallas's Cat walking in the snow, . . .*    *photo of 24 y.o woman, night street, . . .*    *coastline, lighthouse, waves, sunlight, . . .*

**MeinaMix**      **Realistic Vision**      **MoXin**      **Oil painting**

*1girl, white hair, purple eyes, dress, petals, . . .*    *a cyberpunk city street, night time, . . .*    *a bird sits on a branch, ink painting, . . .*    *sunset, orange sky, fishing boats, waves, . . .*

Figure 4: **Qualitative Result.** Each sample corresponds to a distinct personalized T2I. *Best viewed with Acrobat Reader. Click the images to play the animation clips.*

## 5 EXPERIMENTS

We implement AnimateDiff upon Stable Diffusion V1.5 and train motion module using the WebVid-10M (Bain et al., 2021) dataset. Detailed configurations can be found in supplementary materials.

### 5.1 QUALITATIVE RESULTS

**Evaluate on community models.** We evaluated the AnimateDiff with a diverse set of representative personalized T2Is collected from Civitai (2022). These personalized T2Is encompass a wide range of domains, thus serving as a comprehensive benchmark. Since personalized domains in these T2Is only respond to certain "trigger words", we abstain from using common text prompts but refer to the model homepage to construct the evaluation prompts. In Fig. 4, we show eight qualitative results of AnimateDiff. Each sample corresponds to a distinct personalized T2I. In the second row of Figure 1, we present the outcomes obtained by integrating AnimateDiff with MotionLoRA to achieve shot type controls. The last two samples exhibit the composition capability of MotionLoRA, achieved by linearly combining the individually trained weights.

**Compare with baselines.** In the absence of existing methods specifically designed for animating personalized T2Is, we compare our method with two recent works in video generation that can be adapted for this task: **1) Text2Video-Zero** (Khachatryan et al., 2023) and **2) Tune-a-Video** (Wu et al., 2023). We also compare AnimateDiff with two commercial tools: **3) Gen-2 (2023)** for text-to-video generation, and **4) Pika Labs (2023)** for image animation. The results are shown in Fig. 5.

### 5.2 QUANTITATIVE COMPARISON

We conduct the quantitative comparison through user study and CLIP metrics. The comparison focuses on three key aspects: *text alignment*, *domain similarity*, and *motion smoothness*. The results are shown in Table 1. Detailed implementations can be found in supplementary materials.

**User study.** In the user study, we generate animations using all three methods based on the same personalized T2I models. Participants are then asked to individually rank the results based on the

|  |  |  |  |
|---|---|---|---|
| Tune-A-Video | AnimateDiff | T2V-Zero | AnimateDiff |

*a raccoon is playing guitar, soft lighting, . . .*     *a horse galloping on the street, best quality, . . .*

Figure 5: **Qualitative Comparison.** *Best viewed with Acrobat Reader. Click the images to play the animation clips.*

Table 1: Quantitative comparison. A higher score indicates superior performance.

| Method | User Study (↑) | | | CLIP Metric (↑) | | |
|---|---|---|---|---|---|---|
| | Text. | Domain. | Smooth. | Text. | Domain. | Smooth. |
| Text2Video-Zero | 1.620 | **2.620** | 1.560 | 32.04 | 84.84 | 96.57 |
| Tune-a-Video | 2.180 | 1.100 | 1.615 | **35.98** | 80.68 | 97.42 |
| **Ours** | **2.210** | 2.280 | **2.825** | 31.39 | **87.29** | **98.00** |

above three aspects. We use the Average User Ranking (AUR) as a preference metric where a higher score indicates superior performance. Note that the corresponding prompts and images are provided for reference for text alignment and domain similarity evaluation.

**CLIP metric.** We also employed the CLIP (Radford et al., 2021) metric, following the approach taken by previous studies (Wu et al., 2023; Khachatryan et al., 2023). When evaluating domain similarity, it is important to note that the CLIP score was computed between the animation frames and the reference images generated using the personalized T2Is.

### 5.3 ABLATIVE STUDY

**Domain adapter.** To investigate the impact of the domain adapter in AnimateDiff, we conducted a study by adjusting the scaler in the adapter layers during inference, ranging from 1 (full impact) to 0 (complete removal). As illustrated in Figure 6, as the scaler of the adapter decreases, there is an improvement in overall visual quality, accompanied by a reduction in the visual content distribution learned from the video dataset (the watermark in the case of WebVid (Bain et al., 2021)). These results indicate the successful role of the domain adapter in enhancing the visual quality of AnimateDiff by alleviating the motion module from learning the visual distribution gap.

**Motion module design.** We compare our motion module design of the temporal Transformer with its full convolution counterpart, which is motivated by the fact that both designs are widely employed in recent works on video generation. We replace the temporal attention with 1D temporal convolution and ensured that the two model parameters were closely aligned. As depicted in supplementary materials, the convolutional motion module aligns all frames to be identical but does not incorporate any motion compared to the Transformer architecture.

**Efficiency of MotionLoRA.** The efficiency of MotionLoRA in AnimateDiff was examined in terms of *parameter efficiency* and *data efficiency*. Parameter efficiency is crucial for efficient model training and sharing among users, while data efficiency is essential for real-world applications where collecting an adequate number of reference videos for specific motion patterns may be challenging.

To investigate these aspects, we trained multiple MotionLoRA models with varying parameter scales and reference video quantities. In Fig. 7, the first two samples demonstrate that MotionLoRA is capable of learning new camera motions (*e.g.*, zoom-in) with a small parameter scale while maintaining comparable motion quality. Furthermore, even with a modest number of reference videos (*e.g.*, $N = 50$), the model successfully learns the desired motion patterns. However, when the number of reference videos is excessively limited (*e.g.*, $N = 5$), significant degradation in quality is observed,

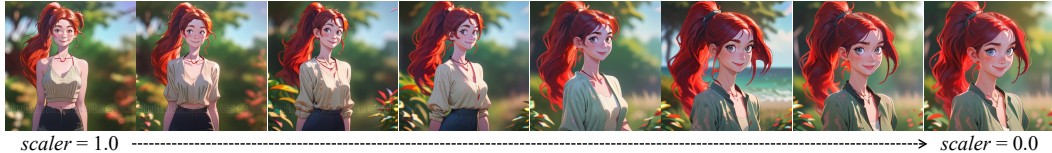

*scaler = 1.0* --------------------------------------------------------------------------------------------------------------------------> *scaler = 0.0*

Figure 6: **Ablation on domain adapter.** We adjust the scaler of the adapter from 1 to 0 to gradually remove its effects. In this figure, we show the first frame of the generated animation.

*rank=2 (∼1M)*      *rank=128 (∼36M)*      *N=5*      *N=50*      *N=1000*

Figure 7: **Ablation on MotionLoRA's efficiency.** *Two samples on the left*: with different network rank; *Three samples on the right*: with different numbers of reference videos. *Best viewed with Acrobat Reader. Click the images to play the animation clips.*

suggesting that MotionLoRA encounters difficulties in learning shared motion patterns and instead relies on capturing texture information from the reference videos.

### 5.4 CONTROLLABLE GENERATION.

The separated learning of visual content and motion priors in AnimateDiff enables the direct application of existing content control approaches for controllable generation. To demonstrate this capability, we combined AnimateDiff with ControlNet (Zhang et al., 2023) to control the generation with extracted depth map sequence. In contrast to recent video editing techniques (Ceylan et al., 2023; Wang et al., 2023a) that employ DDIM (Song et al., 2020) inversion to obtain smoothed latent sequences, we generate animations from randomly sampled noise. As illustrated in Figure 8, our re-

*city street, neon, fog, closeup portrait photo of young woman in dark clothes, …*

Figure 8: **Controllable generation.** *Best viewed with Acrobat Reader. Click the images to play the animation clips.*

sults exhibit meticulous motion details (such as hair and facial expressions) and high visual quality.

## 6 CONCLUSION

In this paper, we present AnimateDiff, a practical pipeline directly turning personalized text-to-image (T2I) models for animation generation once and for all, without compromising quality or losing pre-learned domain knowledge. To accomplish this, we design three component modules in AnimateDiff to learn meaningful motion priors while alleviating visual quality degradation and enabling motion personalization with a lightweight fine-tuning technique named MotionLoRA. Once trained, our motion module can be integrated into other personalized T2Is to generate animated images with natural and coherent motions while remaining faithful to the personalized domain. Extensive evaluation with various personalized T2I models also validates the effectiveness and generalizability of our AnimateDiff and MotionLoRA. Furthermore, we demonstrate the compatibility of our method with existing content-controlling approaches, enabling controllable generation without incurring additional training costs. Overall, AnimateDiff provides an effective baseline for personalized animation and holds significant potential for a wide range of applications.

# 7 ETHICS STATEMENT

We strongly condemn the misuse of generative AI to create content that harms individuals or spreads misinformation. However, we acknowledge the potential for our method to be misused since it primarily focuses on animation and can generate human-related content. It is also important to highlight that our method incorporates personalized text-to-image models developed by other artists. These models may contain inappropriate content and can be used with our method.

To address these concerns, we uphold the highest ethical standards in our research, including adhering to legal frameworks, respecting privacy rights, and encouraging the generation of positive content. Furthermore, we believe that introducing an additional content safety checker, similar to that in Stable Diffusion (Rombach et al., 2022), could potentially resolve this issue.

# 8 REPRODUCIBILITY STATEMENT

We provide comprehensive implementation details for the training and inference of our method in supplementary materials, aiming to enhance the reproducibility of our approach. We also make both the code and pre-trained weights open-sourced to facilitate further investigation and exploration.

## ACKNOWLEDGEMENT

This project is funded in part by Shanghai AI Laboratory (P23KS00020, 2022ZD0160201), CUHK Interdisciplinary AI Research Institute, and the Centre for Perceptual and Interactive Intelligence (CPII) Ltd under the Innovation and Technology Commission (ITC)'s InnoHK.

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

APPENDIX

# A    IMPLEMENTATION DETAILS

**Training.** We utilize the WebVid-10M dataset (Bain et al., 2021), a large-scale video dataset consisting of approximately 10.7 million text-video data pairs to train the motion module. This dataset offers diverse motion categories, which significantly facilitates the learning process of the motion module. We adopt a training resolution of $256 \times 256$ to balance training efficiency and motion quality. To train the domain adapter, we randomly sample static frames and resize them to the target resolution. For the motion module and MotionLoRA, we uniformly sample the videos at a stride of 4 to get video clips at a length of 16. We use a learning rate of $1 \times 10^{-4}$ and train the motion module with 16 NVIDIA A100s for 5 epochs.

**Inference.** As described in the main paper, at inference, we first inflate a personalized text-to-image model and insert the pre-trained motion module to constitute the corresponding animation generator. In our experiment setup, we generate animations at a resolution of $512 \times 512$ using a DDIM (Song et al., 2020) sampler with classifier-free guidance. We referred to the model's official web page to determine the denoising hyperparameters (guidance scale, LoRA scaler, *etc.*) and generally adopted the same settings.

Table 2: Community models for evaluation.

| Model Name | Domain | Type |
|---|---|---|
| ToonYou[1] | 2D Cartoon | T2I Base Model |
| MeinaMix[2] | 2D Anime | T2I Base Model |
| Lyriel[3] | Stylistic | T2I Base Model |
| RCNZ Cartoon 3d[4] | 3D Cartoon | T2I Base Model |
| epiC Realism[5] | Realistic | T2I Base Model |
| Realistic Vision[6] | Realistic | T2I Base Model |
| Oil painting[7] | Stylistic | LoRA |
| MoXin[8] | Stylistic | LoRA |
| TUSUN[9] | Concept | LoRA |

**Models for evaluation.** To ensure a comprehensive benchmark, we selected nine representative personalized T2I models from Civitai (2022), a model-sharing platform that enables artists to upload their creations. As illustrated in Table 2, these models encompass diverse domains such as 2D anime, stylistic painting, and realistic photographic images. They also cover a wide range of subjects, including portraits, animals, landscapes, *etc.*. This selection ensures a comprehensive evaluation of our approach across various domains and subjects.

**Baselines adaptation.** To adapt the two academic baselines for personalized animation generation, we followed the recommended best practices in the respective papers and performed parameter tuning on a case-by-case basis. For Tune-A-Video (Wu et al., 2023), we use a reference video on the project's webpage and fine-tune the network after replacing the T2I backbone with a personalized one, as suggested in the paper. Regarding Text2Video-Zero (Khachatryan et al., 2023), we directly generate video clips upon the personalized T2Is without any modifications. In addition, we conducted qualitative comparisons with two commercial tools for video generation and image animation, namely Gen-2 (2023) and Pika Labs (2023). For Gen2, we employed personalized T2I images as image prompts to generate the corresponding videos. As for Pika Labs, we utilized it to animate still images generated by the personalized T2Is.

---

[1] https://civitai.com/models/30240/toonyou
[2] https://civitai.com/models/7240?modelVersionId=119057
[3] https://civitai.com/models/22922/lyriel
[4] https://civitai.com/models/66347?modelVersionId=82547
[5] https://civitai.com/models/25694?modelVersionId=143906
[6] https://civitai.com/models/4201?modelVersionId=130072
[7] https://civitai.com/models/84542/oil-paintingoil-brush-stroke
[8] https://civitai.com/models/12597/moxin
[9] https://civitai.com/models/33194/leosams-pallass-catmanul-lora

| (SD1.5) Close up of grapes on table. | (SD1.5) Sunset time-lapse at the beach. | (SD1.5) An astronaut flying in space. | (SD1.5) A bigfoot walking. |

Figure 9: **Results on base T2I backbone.** *Best viewed with Acrobat Reader. Click the images to play the animation clips.*

**User study.** To ensure a fair comparison between our method and the baselines, we generated 20 animations for each method without cherry-picking, resulting in 20 sets of triple pairs. Subsequently, we conducted a user study involving ten participants. Each participant is presented with the three samples generated by different methods at one time and asked to rank them based on three specific aspects: *text alignment*, *domain similarity*, and *motion smoothness*. To evaluate text alignment, we provide the corresponding text prompt to the users and request them to rank the samples accordingly. To assess domain similarity, we initially generate reference images using the same personalized T2I. These reference images are then presented to the users, who are asked to rate the animations based on their perceived similarity to the reference images. Regarding motion smoothness, users are instructed to rank the animations based on the consistency of the motion.

**CLIP metric.** Using the generated animations, we first extract the CLIP image embeddings of each frame and then compute the cosine similarity under different settings. To assess text alignment, we computed the average similarity between the prompt embedding and the embeddings of individual frames. For evaluating domain similarity, we computed the CLIP score between the reference images and the frames of the animations. To measure motion smoothness, we calculated the similarity between all pairs of video frames and reported the average number.

## B   ADDITIONAL DISCUSSIONS

### B.1   VISUAL QUALITIES ON BASE T2I.

By integrating the motion module with the base T2I that the motion module is pre-trained upon, *i.e.*, Stable Diffusion V1.5, AnimateDiff demonstrates capabilities in general T2V generation. We showcase such ability by generating videos with commonly used textual prompts in previous works (Zhou et al., 2022a; Blattmann et al., 2023). As illustrated in Fig. 9, without the enhancement from personalized T2I models, the domain of the synthetic videos corresponds closely with the pre-training dataset WebVid-10M Bain et al. (2021).

### B.2   DOMAIN ADAPTER VISUALIZATION

To further validate the effectiveness of the domain adapter, we conduct an additional ablative study where the motion module is trained with the domain adapter entirely removed from the pipeline. We qualitatively compare the personalized T2I animation results upon three baselines: (1) training without adapter; (2) full pipeline with scaler $\alpha$ set to 1; (3) full pipeline with scaler $\alpha$ set to 0.

As shown in Fig. 10, when the domain adapter is completely removed from the training pipeline, visual attributes inherent to the training dataset, specifically watermarks, emerge in the synthetic animations (1st row). This arises due to the intertwining of visual appearance and motion learning during the motion module's training phase, resulting in the watermark pattern being learned by the motion module and subsequently transferred to other personalized T2I backbones. Similarly, watermarks appear when the adapter exerts its full impact (2nd row). In contrast, by fitting the visual distribution to a separate domain adapter and eliminating it during inference, our full pipeline

*train w/o adapter*        *full pipeline,* $\alpha = 1$        *full pipeline,* $\alpha = 0$

Figure 10: **Abaltions on domain adapter.** *Best viewed with Acrobat Reader. Click the images to play the animation clips.*

*w/o scale-up training*        *w/ scale-up training*        *w/o scale-up training*        *w/ scale-up training*

Figure 11: **Ablations on scale-up training.** *Best viewed with Acrobat Reader. Click the images to play the animation clips.*

(3rd row) achieves superior quality devoid of watermarks. This implies that the visual distribution within the training dataset can be effectively eliminated by merely dropping the adapter.

### B.3 BENEFITS FROM SCALE-UP TRAINING

In practice, we find that the overall quality of the generated animations benefits from scale-up training. This involves training with larger batch sizes, video resolution, and the number of total optimizing iterations. In Fig. 11, we present two pairs of qualitative comparisons between motion modules trained with standard and scale-up training. Under the scale-up training setting, we train the motion module on the resolutions of $320 \times 512$, with $8\times$ larger batch size compared to the standard setting. The result indicates that considerable enhancement in motion amplitude and diversity can be achieved through an increase in the training scale. For instance, the camera involves view angle changes (2nd row) in contrast to mere zooming (1st row). The character's head displays turning movements (4th row) rather than solely facing forward (3rd row).

## C LIMITATIONS

### C.1 MOTION PRIORS IN ANIMATEDIFF

The transferable motion priors in AnimateDiff are learned from a large-scale video dataset WebVid-10M (Bain et al., 2021) that encompasses mainly real-world footage. Supported by the richness and diversity of the dataset, the motion module can learn real-world motions (Ding et al., 2022) like sea waves, vehicular movement, and human actions, which are modeled by the temporal self-attention mechanism. Therefore, the motion priors largely depend on the dataset coverage and accuracy, which introduces potential limitations discussed as follows.

**Motion diversity and complexity.** In practice, we find that the motion module pre-trained on WebVid-10M performs well on non-violent motions such as fluid (*e.g.*, ocean waves, fog, etc.), rigid objects (*e.g.*, cars, boats), and simple human movements (*e.g.*, walking, facial expressions). This aligns with our observation that the training dataset predominantly encompasses these motions.

*Stable Diffusion V1.5*        *Stable Diffusion XL*

Figure 12: **Improvements on the more powerful backbone.**    *Best viewed with Acrobat Reader. Click the images to play the animation clips.*

However, the module struggles with complex motions that are infrequent in the training dataset and challenging to represent via short video clips during training, e.g., dance movements and drastic scene changes. These instructions often result in static synthetic outcomes or unnatural deformations. Potential solutions could involve enriching the training set's motion diversity or training with larger resolution and extended clip length, which will help to better model motion patterns.

**Text-motion alignment.** In the pre-training dataset, WebVid-10M, most text labels primarily describe visual content while overlooking detailed motion descriptions. Consequently, this leads to the animations generated by AnimateDiff exhibiting little response to the motion descriptions. Notwithstanding, this phenomenon *does not* imply that the motion module does not acquire corresponding motion priors. For instance, zoom-in/out effects frequently appear in the pre-training videos. However, their text labels typically contain only broad "zooming" tags, making it difficult for the motion module to distinguish the difference between zoom-in and zoom-out accurately. As a result, utilizing a "zoom in" prefix alongside the common text prompt generates both zoom-in and zoom-out effects, indicating the need for a video dataset with more accurately labeled motion tags. This also suggests that MotionLoRA does not learn new motion patterns entirely from scratch but refines and enhances the pre-existing motion priors (regardless of whether they can be triggered by text) obtained during pre-training, enabling the motion module to express such priors as desired during inference.

## C.2 DEPENDENCY ON IMAGE BACKBONE

Under the decoupled training strategy, the motion and visual content in the generated animations originate from the pre-trained motion module and the underlying image backbone, respectively. Consequently, the performance of the entire pipeline of AnimateDiff is heavily reliant on the underlying T2I models. If the base model struggles to respond appropriately to the text prompt and fails to generate accurate content, the additional motion module is unlikely to compensate for this weakness. Conversely, superior image backbones can enhance the synthetic results. To demonstrate this, we implement AnimateDiff on Stable Diffusion XL (Podell et al., 2023) and compare general T2V results on rare semantic compositions against the Stable Diffusion V1.5 version, as depicted in Fig. 12. The figure illustrates that the synthetic video based on SDXL achieves better visual composition and semantic alignment.

Practically, to mitigate potential limitations introduced by the foundational T2I models, employing off-the-shelf modules such as IP-adapter (Ye et al., 2023) for additional style/content reference, ControlNet (Zhang et al., 2023) for spatial composition corrections, could be beneficial.

## D  MORE VISUAL RESULTS

In Fig. 13, we show more visual results of AnimateDiff and the results of further combing AnimateDiff with MotionLoRA to achieve shot type control. In Fig. 14, we show more qualitative comparisons between AnimateDiff and four academic and commercial baselines. In Fig. 15, we compare two motion module architectures, *i.e.*, the full convolution one and its Transformer counterpart.

*Oil painting*     *Realistic Vision*     *Realistic Vision*     *Realistic Vision*

*city, rainy day, wet, car, a bustling street, oil painting, . . .*

*photo of 18 y.o woman in dress, night city street, motion blur, . . .*

*photo of a cyberpunk city street, night time, dark atmosphere, . . .*

*b&w photo of 42 y.o man in black clothes, bald, face, half body, . . .*

*Oil painting*     *Lyriel*     *Oil painting*     *epiC Realism*

*oil painting, black pearl pirate ship, wind, waves, night time, . . .*

*portrait of halo, sunglasses, blue eyes, tartan scarf, . . .*

*oil painting, mountain, lake water, boat, forest, masterpiece, . . .*

*landscape, a rocky mountain with milky way, nighttime, . . .*

*epiC Realism*     *epiC Realism*     *ToonYou*     *Realistic Vision*

*(zoom-in) A nebula in universe, highly detailed, colorful, . . .*

*(rolling) landscape of a aesthetically Belgium and wildflower, . . .*

*(rolling) 1boy, dark skin, playing guitar, concert, stage lights, . . .*

*(zoom-in) cabins in the forest, water, aurora in the sky, fog, . . .*

*Oil Painting*     *MoXin*     *RCNZ Cartoon 3d*     *Lyriel*

*(panning) oil painting, house, grass, wheat field laboring crowd, . . .*

*(panning) fantastic composition, old Chinese town, . . .*

*(panning) a golden labrador, warm vibrant colours, . . .*

*(tilting) waters, canyon, sunlight, traveler, high quality, . . .*

Figure 13: **Additional qualitative results.** *Best viewed with Acrobat Reader. Click the images to play the animation clips.*

Tune-A-Video AnimateDiff T2V-Zero AnimateDiff

*a man is playing guitar, dramatic lighting, . . .*      *a girl is playing guitar, wavy hair, upper body, . . .*

Pika Labs (2023) AnimateDiff Gen-2 (2023) AnimateDiff

*cabins in the forest, water, aurora in the sky, fog, . . .*      *taxi, rear view, New York city at night, . . .*

Pika Labs (2023) AnimateDiff Gen-2 (2023) AnimateDiff

*sunset, orange sky, fishing boats, ocean waves, . . .*      *a woman standing on the road at night, . . .*

Figure 14: **Qualitative comparison.** *Best viewed with Acrobat Reader. Click the images to play the animation clips.*

*Convolution*      *Transformer*      *Convolution*      *Transformer*

Figure 15: Module design comparison.

