# OpenReview forum: "AnimateDiff: Animate Your Personalized Text-to-Image Diffusion Models without Specific Tuning"
_ICLR.cc/2024/Conference — ICLR 2024 spotlight_

### Official Review · Reviewer_eMMP · 2023-10-29

**Soundness:** 3 good
**Presentation:** 3 good
**Contribution:** 2 fair
**Rating:** 6
**Confidence:** 3

**Summary:**

This paper proposes a practical plug-in-play solution for a personalized T2I video generation model. It consists of three parts, the first pipeline uses lora layer to adapt domain from image pre-training to video training domain due to visual quality and motion blur issues. The second part is a motion module trying to learn the motion prior with a temporal transformer model. The last part is optional, which is a motion-lora design to adapt to a personalized video domain. At inference time, the general-purpose motion module can be used in any pre-trained personalized T2I model to animal the image model. Overall, this paper is well-written and the solution is very practical. Empirically, the method has competitive performance compared to other general-purpose text-to-video models.

**Strengths:**

– This paper is trying to solve a novel and practical problem in the text-to-video generation domain. How to turn a pre-trained personalized T2I model into a video model without training is attractive and practically useful.
– Qualitatively, this method shows impressive result and demonstrate the generalization ability of the motion module to different T2I model
– It also has an extensive ablation study showing the effectiveness of the domain adapter and motion module design choices.

**Weaknesses:**

– Lack of technical novelty, the proposed domain adapter is based on LoRA and the motion module is following standard feature inflation and the architecture is following timesformer.

**Questions:**

-- I do have some concerns about the generalization of the motion module. It is not clear to me how reliable to use LoRA for domain adaption, and if the motion module does not generalize well, should we further fine-tune the motion module on the task domain data?

---

> ### Author Response · Authors · 2023-11-21
> **Response to Reviewer eMMP**
>
> **Q1: Lack of technical novelty, the proposed domain adapter is based on LoRA and the motion module is following standard feature inflation and the architecture is following timesformer.**
>
> As motivated in Sec. 1, our primary goal is to develop an effective pipeline to animate personalized T2I models. The technical novelty of our work is not in the modules themselves, but in the unique way they are combined and utilized, leading to an innovative solution that is greater than simply putting them together.
>
> Since our goal is to turn a T2I model into an animation generator, feature inflation is the most efficient practice. Extensive experiments in Sec. 5 and high-quality visual results in Fig. 1,4,9,10 also validate the effectiveness of our architecture choices.
>
> Additionally, it is not ad-hoc to use these two modules. In Sec. 5.3 and Sec. 4.2, we investigate different options for the motion module, e.g., a convolution version, and it turns out the current simple design is strong and efficient. In terms of the domain adapter, as discussed in Sec. 4.1 and Sec. 5.3, it’s worth noting that we didn’t use it for domain personalization as in previous work. Instead, we only use it during training to reduce the gap between video training data and high-quality training images of T2I. This not only results in high-quality video generation but also offers a new usage of domain adapters.
>
> **Q2: I do have some concerns about the generalization of the motion module. It is not clear to me how reliable to use LoRA for domain adaption, and if the motion module does not generalize well, should we further fine-tune the motion module on the task domain data?**
>
> To be clear, the domain adapter (implemented as an image LoRA) aims to bridge the gap between the training videos and the images (for base T2I models), which would be discarded after training. Sec.5.3 and Fig.6 demonstrate that this design indeed helps alleviate the negative effects of the video dataset.
>
> The motion module is proposed to learn the motion pattern regarding the video dataset. For instance, videos in the training set usually involve how humans should act and how cars should move. In this case, the generalization ability largely depends on the motion that the training set covers. After this learning, personalized video generation is enabled with a user-specific domain by inserting a personalized image LoRA.
>
> One straightforward solution for some specific complex movements that might not appear in the training set is to collect the larger-scale corresponding videos into the training set and tune the motion module. However, this is costly and impractical. We, therefore, propose MotionLoRA as a complementary method that could learn a specific motion pattern based on the general motion prior (learned by our motion module). As mentioned in Sec. 4.3, training such a MotionLoRA usually requires 20-50 reference videos and takes around 1 hour, which provides a more practical solution.
>
> Combining them all, our AnimateDiff could not only alleviate the effect of training videos but also capture their motion prior. More importantly, it could quickly and efficiently adapt to other new/complex motion patterns that the aforementioned training set cannot cover.

---

> ### Author Response · Authors · 2023-11-22
> **Any further comments?**
>
> Dear reviewer,
>
> We kindly request your feedback on whether our response has addressed your concerns. If you have any remaining questions or concerns, we are happy to address them. Thank you for your time and consideration!

---

### Official Review · Reviewer_ih9K · 2023-10-30

**Soundness:** 3 good
**Presentation:** 4 excellent
**Contribution:** 3 good
**Rating:** 8
**Confidence:** 5

**Summary:**

This paper introduces "AnimateDiff," a revolutionary framework designed to infuse motion dynamics into personalized text-to-image (T2I) diffusion models without necessitating model-specific tuning. AnimateDiff features a versatile, plug-and-play motion module at its core, enabling the generation of animated images. The motion module is trained to adaptively learn and apply transferable motion priors from real-world videos, allowing for seamless integration into various personalized T2Is, and fostering the creation of a personalized animation generator.

Additionally, the paper unveils "MotionLoRA," a novel, lightweight fine-tuning technique engineered for AnimateDiff. MotionLoRA facilitates the adaptation of pre-trained motion modules to new motion patterns, such as diverse shot types, ensuring adaptability with minimal training and data collection efforts.

**Strengths:**

1. AnimateDiff empowers personalized text-to-image (T2I) models with animation generation capabilities, eliminating the need for specific fine-tuning. Besides, it demonstrates the efficacy of Transformer architectures in modeling motion priors, contributing valuable insights to the field of video generation.

2. The authors propose "MotionLoRA," an ingenious, lightweight fine-tuning technique enabling pre-trained motion modules to adapt to new motion patterns seamlessly.

3. AnimateDiff and MotionLoRA are rigorously tested against representative community models, academic baselines, like Gen2 and Tune-A-Video.

**Weaknesses:**

This paper is well-composed and presents a notable contribution to the field of text-to-image (T2I) diffusion models.

However, one aspect that could be further refined or discussed is the dependency of AnimateDiff on the underlying T2I models. It appears that the success of AnimateDiff is inherently tied to the performance and reliability of the existing T2I models it seeks to enhance. Specifically, the methodology might face challenges if the base T2I models struggle or fail to accurately generate images based on user-provided prompts. In such cases, AnimateDiff might encounter difficulties in achieving its animation objectives, potentially limiting the overall effectiveness and applicability of the proposed solution.

It might be beneficial for the paper to address this dependency, possibly discussing strategies or considerations to mitigate potential challenges arising from limitations in the foundational T2I models. This would strengthen the robustness and adaptability of AnimateDiff, ensuring its broader success and applicability in various T2I contexts and scenarios.

**Questions:**

### 1. Domain Adapter Visualization:
Could the authors provide visualization results that specifically exclude the domain adapter training state? To clarify, I am interested in viewing the outcomes when the domain adapter training (stage 1) is entirely omitted from the process, not just when the parameter $\alpha$ is set to zero at the inference state.

### 2. Visual Quality of AnimateDiff in Base T2I Models:

Regarding the visual quality, how does AnimateDiff perform when applied directly to the base T2I model that you used to train the motion module on WebViv? I am particularly interested in understanding the performance and visual outcomes of AnimateDiff when integrated with the foundational T2I models, excluding any enhancements or personalizations from external models such as Civitai.

---

> ### Author Response · Authors · 2023-11-21
> **Response to Reviewer ih9K**
>
> **Q1: About the dependency of AnimateDiff on the underlying T2I models.**
>
> Current AnimateDiff indeed relies on the underlying T2I models. Due to such dependency, our method could inherit their appealing generated appearance and might hardly avoid some failure cases caused by T2I models. On the other hand, AnimateDiff could be further improved with the advancement of T2I models. We implement AnimateDiff on Stable Diffusion XL (SDXL), which enhances the visual quality and text conditioning on T2I. The comparison of two different T2I models is presented in Appendix D.2 and Fig. 20,24, showing that our AnimateDiff could also be compatible with other advanced T2I models, namely, avoid struggling with the limitations caused by a certain T2I model. We have already updated the related discussion in Appendix D.
>
> **Q2: Domain Adapter Visualization.**
>
> We retrain the motion module with the adapter entirely removed from the pipeline and present the visualizations in Appendix C.2 and Fig.18,22. Obviously, when the domain adapter is entirely removed from the training pipeline, the visual appearance in the training dataset, i.e., the watermarks, appears on the synthetic animations (1st row). Similarly, the watermarks appear when the adapter has its full impact (2nd row). In contrast, by fitting the visual distribution to a separate domain adapter and dropping it at inference, our full pipeline (3rd row) achieves favorable quality with no watermarks.
>
> **Q3: Visual Quality of AnimateDiff in Base T2I Models.**
>
> Although animating the original base T2I model is not our focus, we present the generated videos upon the original T2I in Appendix C.1 and Fig.17,21. It is obvious that the synthesized frames *remain smooth over time*. However, due to the lack of enhancement from personalized T2I models, the produced image corresponds closely with the video dataset (containing watermarks sometimes). Appendix D.2 and Fig.20,24 showcase the results from enhanced base T2I models (like SDXL). Since it is challenging to collect a larger-scale and higher-quality dataset than WebVid, we recommend users leverage the power of personalized image models since it is obviously more flexible and practical.

---

> ### Author Response · Authors · 2023-11-22
> **Any further comments?**
>
> Dear reviewer,
>
> We kindly request your feedback on whether our response has addressed your concerns. If you have any remaining questions or concerns, we are happy to address them. Thank you for your time and consideration!

---

### Official Review · Reviewer_31Uu · 2023-10-31

**Soundness:** 4 excellent
**Presentation:** 4 excellent
**Contribution:** 4 excellent
**Rating:** 8
**Confidence:** 4

**Summary:**

The paper introduces AnimateDiff, a novel pipeline designed to convert static Text-to-Image (T2I) models into animation generators without the need for model-specific fine-tuning. The process uses a plug-and-play motion module named MotionLoRA that learns motion priors from video datasets and can be integrated directly into personalized T2Is to produce smooth animations. Training involves fine-tuning a domain adapter on the base T2I, introducing a motion module, and then adapting this pre-trained module to specific motion patterns using Low-Rank Adaptation (LoRA). Evaluation was performed on various T2I models yielding promising results, demonstrating that a Transformer architecture is effective for capturing appropriate motion priors.

**Strengths:**

- The proposed concept offers a robust and user-friendly plugin to embed motion priors into general T2I models. The model designs are sensible and meaningful. For instance, the domain adapter plays a critical role in mitigating adverse impacts from training data, and MotionLoRA efficiently adjusts to new motion patterns.

- The experiments demonstrate promising applications, including the animation of diverse-style T2I models and controllable dynamic generation using ControlNet.

**Weaknesses:**

- In certain animation results, the motion amplitudes are minimal and flickering between frames is noticeable. This suggests room for improvement in the motion prior. These issues may be related to the size of the video dataset and model design, and a thorough analysis of these factors should be included in the limitations section.

- Qualitative comparisons are not enough. Please provide visual results for more cases like in Figure 5.

- For image animation, the identity preservation is poor, in other words, the characters in the animation results are not very similar to the one in the input image.

**Questions:**

Why does the scale of the domain adaptor in Figure 6 appear to be visually linked with camera movement?

---

> ### Author Response · Authors · 2023-11-21
> **Response to Reviewer 31Uu**
>
> **Q1: In certain animation results, the motion amplitudes are minimal, and flickering between frames is noticeable. A thorough analysis of these factors should be included in the limitations section.**
>
> Thanks for pointing it out. You are right about the future improvement of motion prior. Our experimental results in Appendix C.3 and Fig. 19,23 demonstrate that the generated motions could benefit from *scale-up training*, producing videos with better motion amplitude, diversity, and temporal consistency. This also implies that the current model design is *capable enough* of capturing larger motion. In addition, enlarging the video-text paired dataset is indeed a potential avenue for further advancing our approach. However, it is important to acknowledge that this task can be challenging and costly at this moment. We have already updated the related discussion in Appendix D.
>
> **Q2: Qualitative comparisons are not enough. Please provide visual results for more cases like in Figure 5.**
>
> We have updated more qualitative results in Fig.9,10,11 on pages 16,17. Importantly, 60 samples are involved in our user study in Tab.1 for quantitative comparison, demonstrating the effectiveness of our approach.
>
> **Q3: For image animation, the identity preservation is poor, in other words, the characters in the animation results are not very similar to the one in the input image.**
>
> There might be a need for clarification since our method aims to generate personalized videos given a text prompt. Although animating a given image might be achieved in a certain way, it is *not very related* to this submission. Overall, the temporal consistency could also be improved through scale-up training, as shown in  Appendix C.3 and Fig.19,23.
>
> **Q4: Why does the scale of the domain adaptor in Figure 6 appear to be visually linked with camera movement?**
>
> There might be a misunderstanding. In terms of the domain adapter, it’s worth noting that we didn’t use it for domain transfer during inference as in previous work. Instead, we only use it during training to reduce the gap between video training data and high-quality training images of T2I. This not only results in high-quality video generation but also offers a new usage of domain adapters.
>
> The pose change in Fig. 6 is not due to the camera movement but because of the varying image distribution. When adjusting the scale of the domain adaptor from 1.0 to 0.0, the generated appearance goes to the personalized image domain from the WebVid domain.

---

> ### Author Response · Authors · 2023-11-22
> **Any further comments?**
>
> Dear reviewer,
>
> We kindly request your feedback on whether our response has addressed your concerns. If you have any remaining questions or concerns, we are happy to address them. Thank you for your time and consideration!

---

### Official Review · Reviewer_Jp9x · 2023-11-02

**Soundness:** 3 good
**Presentation:** 4 excellent
**Contribution:** 3 good
**Rating:** 6
**Confidence:** 4

**Summary:**

The paper introduces a simple method for generating animations using a customized text-to-image model adapted for video. It innovates by transforming a text-to-image model to produce a sequence of frames, simulating animation. This is achieved through frame-by-frame processing with original diffusion model and the integration of a transformer-based motion module that processes temporal information across patches from all frames. A domain adapter is introduced to overcome the challenge of dataset-specific artifacts, enhancing the quality of the video output.  In the end, the authors also demonstrate the model's ability to efficiently learn new motion given the unified representation including zoom in and rolling.  Quantitative and qualitative comparisons with  a broad spectrum of existing models validated the effectiveness of the proposed approach.

**Strengths:**

S1: The paper's foremost strength lies in its elegant balance between simplicity and performance. With only minor change to the base stable diffusion model—principally, the addition of a few transformer layers—the method yields robust quantitative outcomes, positioning it as a strong baseline for text-to-video synthesis within the research community.

S2: The simplicity of the approach belies its technical depth. The integration of new domain adaptation layers effectively mitigates artifact issues typical in video datasets. Furthermore, the use of LORA for streamlined learning of new motions, coupled with transformer layers for temporal data synthesis, represents straightforward yet impactful innovations, all substantiated by thorough ablation studies.

S3: The paper presents good qualitative and quantitative results.

S4: The paper is well written. It combines clear, concise text with instructive visuals, making the methodology and results accessible and understandable."

**Weaknesses:**

W1: Some modules are not that novel from a technical perspective. E.g. the transformer block is also explored in GEN-2 [1]. LORA and adapter are also commonly used in various personalization papers.

W2: The paper somewhat overlooks a thorough discussion of the method's limitations. Specifically, it would be beneficial for the authors to clarify the learning dynamics of the motion module when pre-trained on the WebVid dataset—whether motions like zoom-ins and rolls require explicit training (using LORA with curated dataset) or can be inferred from textual cues alone. Additionally, a systematic categorization of which motions are effectively learned and which are not could significantly enhance the reader's understanding of the model’s practical applications and boundaries.


[1] Esser, Patrick, et al. "Structure and content-guided video synthesis with diffusion models." Proceedings of the IEEE/CVF International Conference on Computer Vision. 2023.

**Questions:**

Please see my comments in the weakness section.

---

> ### Author Response · Authors · 2023-11-21
> **Response to Reviewer Jp9x**
>
> **Q1: Some modules are not that novel from a technical perspective. E.g., the transformer block is also explored in GEN-2. LORA and adapter are also commonly used in various personalization papers.**
>
> As motivated in Sec. 1, our primary goal is to develop an effective pipeline to animate personalized T2I models. The technical novelty of our work is not in the modules themselves, but in the unique way they are combined and utilized, leading to an innovative solution that is greater than simply putting them together.
>
> And it is not ad-hoc to use these two modules. In Sec. 5.3 and Sec. 4.2, we investigate different options for the motion module, e.g., a convolution version, and it turns out the current simple design is strong and efficient. In terms of the domain adapter, as discussed in Sec. 4.1 and Sec. 5.3, it's worth noting that we didn't use it for domain personalization as in previous work. Instead, we only use it during training to reduce the gap between video training data and high-quality training images of T2I. This not only results in high-quality video generation but also offers a new usage of domain adapters.
>
> **Q2: The paper somewhat overlooks a thorough discussion of the method's limitations.**
>
> We have added a section in the revised submission to discuss the limitations of our method (see pages 22-23, Appendix D). We address the concerns in [W2] in the following Q3-Q5.
>
> **Q3: About the learning dynamics of the motion module when pre-trained on the WebVid dataset.**
>
> AnimateDiff utilizes data priors to learn motions in the training videos dataset WebVid-10M, which contains mainly real-world footage such as outdoor scenes and human activities. By training upon it, the motion module could learn common real-world motion patterns, e.g., ocean waves, the movement of cars and boats, and human body/hand motions, as shown in Fig. 1. We have already updated the related discussion in Appendix D.1.
>
> **Q4: About the learning of MotionLoRA and whether the motion patterns can be inferred from text clues.**
>
> In AnimateDiff, whether a kind of motion can be triggered via text prompt depends on whether the training dataset accurately labeled such motion. We investigate the WebVid-10M dataset and find its motion labels very coarse. For instance, the zoom-in and zoom-out videos in WebVid are mostly labeled with a single tag, “zooming.” As a result, using text hints such as “zoom in” results in both zoom-in and zoom-out generation. On the other hand, more complex motion patterns such as “Rolling” do not have a specific text label, and thus, these motion patterns cannot be triggered via simple textual clues.
>
> This observation suggests that MotionLoRA does not learn new motion patterns entirely from scratch but refines and enhances the pre-existing motion priors (regardless of whether they can be triggered by text) obtained during pre-training. It thus enables the motion module to express the desired motion priors at inference. We have already updated the related discussion in Appendix D.1.
>
> **Q5: About the performance on different motion categories.**
>
> In practice, we find the pipeline works well on non-violent motions such as fluid (e.g., ocean waves, fog, etc.), rigid objects (e.g., cars, boats), and simple human movements (e.g., walking, facial expressions), as shown in Fig. 1,4,9. However, if the pipeline is instructed to generate complex or rare motions rarely seen in the training dataset, e.g., dance or camera movements with drastic scene change, the model will likely generate static animation or unnatural deformations. We have already updated the related discussion in Appendix D.1.

---

> ### Author Response · Authors · 2023-11-22
> **Any further comments?**
>
> Dear reviewer,
>
> We kindly request your feedback on whether our response has addressed your concerns. If you have any remaining questions or concerns, we are happy to address them. Thank you for your time and consideration!

---

> > ### Comment · Reviewer_Jp9x · 2023-11-22
> > **Reply to Authors**
> >
> > Thank you for the response. They addressed my concerns and I will keep my "acceptance" rating.

---

### Meta-Review · Area_Chair_qzzF · 2023-12-07

**Metareview:**

This work proposes to adapt image generating stable diffusion to output temporally coherent video frames. All the reviewers lean towards accepting the work. Reviewers appreciated the well-written paper with good video generation results. Multiple reviewers raised concerns with respect to technical novelty, but in the end appreciated the high-quality results with simple technique. The reviewers did raise some valuable concerns that should be addressed in the final camera-ready version of the paper, which include adding the relevant rebuttal discussions and revisions in the main paper. The authors are encouraged to make the necessary changes to the best of their ability.

**Justification For Why Not Higher Score:**

The technical novelty is somewhat limited.

**Justification For Why Not Lower Score:**

The results are quite good and this technique is quite effective in adapting image generation models for video generation.

---

### Decision · Program_Chairs · 2024-01-16

Accept (spotlight)